# DAMEX: Dataset-aware Mixture-of-Experts for visual understanding of mixture-of-datasets

Yash Jain[1]     Harkirat Behl[2]     Zsolt Kira[1]     Vibhav Vineet[2]

[1]Georgia Institute of Technology     [2]Microsoft Research

## Abstract

Construction of a universal detector poses a crucial question: How can we most effectively train a model on a large mixture of datasets? The answer lies in learning dataset-specific features and ensembling their knowledge but do all this in a single model. Previous methods achieve this by having separate detection heads on a common backbone but that results in a significant increase in parameters. In this work, we present Mixture-of-Experts as a solution, highlighting that MoEs are much more than a scalability tool. We propose Dataset-Aware Mixture-of-Experts, DAMEX where we train the experts to become an *'expert'* of a dataset by learning to route each dataset tokens to its mapped expert. Experiments on Universal Object-Detection Benchmark show that we outperform the existing *state-of-the-art* by average +10.2 AP score and improve over our non-MoE baseline by average +2.0 AP score. We also observe consistent gains while mixing datasets with (1) limited availability, (2) disparate domains and (3) divergent label sets. Further, we qualitatively show that DAMEX is robust against expert representation collapse. Code is available at `https://github.com/jinga-lala/DAMEX`.

## 1    Introduction

Human visual perception has naturally evolved to excel at recognizing and localizing objects in a wide range of real-world scenarios [25, 33]. However, the field of computer vision has traditionally concentrated on training and evaluating object detection models on specific datasets [42, 3, 43, 7, 20, 28, 34]. These models have shown limited adaptability to new environments [1, 32], leading to decreased accuracy and practicality in real-world settings [30, 27]. Recognizing the challenges posed by the single dataset models and the availability of diverse datasets collected over time, unified models are being developed that merge information from multiple datasets into a single framework [35, 45, 11, 41, 24, 15].

The process of combining datasets presents several challenges that need to be addressed. Firstly, these datasets have been collected over time [5] and for different purposes, resulting in variations across domains. For example, one dataset may contain images from indoor environments [9], while another may be focused on the medical domain [38]. Secondly, imbalances in training images and tail distributions may exist across the datasets. This means that certain datasets and classes might have a substantial number of training examples, while others have only a few training examples. To illustrate, the popular Universal Object-Detection Benchmark (UODB) dataset for robust object detection demonstrates this diversity: the Clipart [13] data subset comprises 500 training images, whereas the COCO dataset [22] includes 35,000 training images. Figure 1 shows such diversities in the real world data, encompassing various aspects such as domains, the number of training images, camera views, scale and label set, taken from the multiple data sources. Can a universal object detection method be designed that effectively addresses the above issues associated with mixing multiple datasets?

37th Conference on Neural Information Processing Systems (NeurIPS 2023).

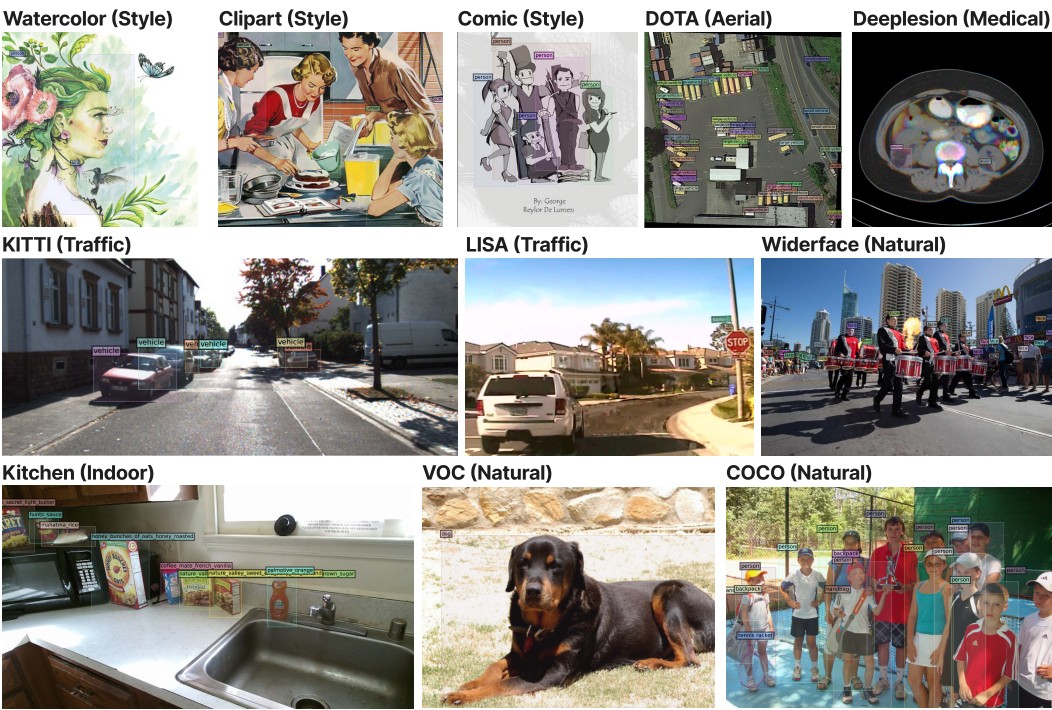

Figure 1: **Samples from Universal Object Detection Benchmark (UODB):** UODB set comprises of a group of 11 datasets which vary in domains, labels, camera view and scale.

To tackle multi-datasets based object detection problem, several approaches have been proposed. [16] manually unifies taxonomy of 7 semantic segmentation datasets using Mechanical Turk whereas our objective is to learn a single universal detector on any mixture of datasets. [45] provides a way to combine multiple separate detectors by learning a unified label space for multiple datasets. Their work proposed a new loss term to unify the label space but requires training separate partitioned detectors heads which are later unified with a drop in performance. [35] trains a universal detector on 11 datasets of Universal Object Detection Benchmark (UODB) and gain robustness by joining different sources of supervision. However, their method requires knowledge of the input dataset domain during test time whereas our objective universal detector should be agnostic to the input data domain making the problem more challenging.

In this work, we propose a Mixture-of-Experts (MoE) [14] based object detection method that can handle all the desired properties of a universal detector. Traditionally, MoEs have been used as a scaling tool in Natural Language Processing [6, 18, 31, 21, 4, 2] and image classification problem [12, 29]. However, we propose that Mixture-of-Experts are more than just scalable learners, instead MoEs are an effective and efficient solution to build a universal model for mixture of datasets. Building over vanilla MoE, we introduce a novel Dataset-aware Mixture-of-Experts model, DAMEX which learns to disentangle dataset-specific features at the MoE layer while pooling information at non-MoE layers. DAMEX learns to route tokens to their corresponding expert so that during inference it automatically chooses the best pathway for the test image through the network. We use DINO [42] based detection architecture for developing our approach.

We perform our evaluation on UODB dataset which comprises of 11 diverse datasets as shown in Figure 1. We compare against previously reported *state-of-the-art* on UODB [35] and recent multi-dataset object-detection method [45]. Moreover, we also compare our method against individual dataset trained DINO detectors along with complete UODB trained DINO model. We report an average +10.2 AP score improvement over previous baseline and a +2.0 AP gain over non-MoE based method. Additionally, we mix dataset pairs that highlight that MoE based detectors are better

in handling, (1) Mixing with limited data availability, (2) Domain Adaptation, (3) Mixing divergent label sets within same domain, and show consistent gains on DAMEX. We qualitatively show that DAMEX results in better expert utilization and avoid representation collapse which is a common problem with MoE training [2].

Our contributions can be summarized as:

- We introduce a novel Dataset-aware Mixture-of-Experts layer, DAMEX which disentangles dataset-specific features within the MoE layer, effectively tackling mixing of heterogeneous datasets. We experimentally show that DAMEX facilitates superior expert utilization and avoid the common issue of representation collapse in vanilla MoE.

- Compared to the baselines, DAMEX does not require test-time dataset labels as it learns to route appropriate input dataset to its corresponding expert during training itself, which is a more challenging setup.

- To the best of our knowledge, we are the first work to explore the potential of MoE as being more than a scalability tool but an effective learner for a mixture of datasets by serving as a knowledge-ensembling strategy within the model architecture with a marginal increase in number of parameters *wrt.* dense architecture.

- We establish a new *state-of-the-art* on UODB set by beating previously reported baseline by an average of +10.2 AP score and outperforming non-MoE baseline by +2.0 AP score on average. Further, we observe consistent improvements in various dataset mixing scenarios, like (1) Limited data availability, (2) Disparate domains, and (3) Divergent label sets.

## 2   Related Work

### 2.1   Mixing Multiple Datasets in Object-detection

Training on multiple datasets has emerged as a robust way to improve task performance of a deep learning model as shown in stereo matching [39], depth estimation [17, 10] and detection performance [35, 16, 45]. Previously, [16] manually unifies taxonomy of 7 semantic segmentation datasets using Mechanical Turk whereas [44] manually merge the label mapping of datasets at the detection head. However, our objective is to build a universal object-detection model which should be able to adapt to any mixture of detection datasets. [37] trains a partitioned detector on three datasets and combine their information through an inter-dataset graph-based attention module. [45] goes a step forward and provides a way to combine multiple separate detectors by learning a unified label space for multiple datasets. Their work proposed a new loss term to unify the label space but requires training separate partitioned detectors heads which are later unified with a drop in performance. Our approach, on the other hand does not rely on unification of labels and does not need specialized partitioned detectors as it utilizes the advantages of a transformer based end-to-end detection pipeline and simply increases the number of classes in the last layer, resulting in marginal increase in parameters. Similar to [37], [35] trains a universal detector on 11 datasets of Universal Object Detection Benchmark (UODB) by having separate detection heads for each dataset with a domain attention module conditioned on datasets for supervision of backbone during training. However, like [37] their method also requires knowledge of the input dataset domain during test time whereas our objective universal detector is agnostic to the input data domain making the detection more challenging.

### 2.2   Mixture-of-Experts

Mixture-of-experts (MoE) [14] is a machine learning concept that employs multiple expert layers, each of which specializes in solving a specific subtask. The experts then work together to solve the entire task at hand. Recently, MoE has been widely applied to large-scale distributed Deep Learning models by using a cross-GPU layer that exchanges hidden features from different GPUs [12, 6, 29, 21, 4, 18, 31]. The MoE approach is differentiated from existing scale-up approaches for DNNs, such as increasing the depth or width of DNNs, in terms of its high cost-efficiency. Specifically, adding more model parameters (experts) in MoE layers does not increase the computational cost per token. Thus, MoE has been studied for scaling for the models of trillion-size parameters in NLP [6, 21, 4, 18].

In machine vision, [12, 29] primarily showed benefits of MoE on ImageNet classification task. In this work, we further pursue this direction by modelling MoE on object detection task and highlight the effectiveness, and robustness of MoEs in learning from a mixture of datasets. Moreover, we improve over expert utilization problem of MoEs [2] in a mixture of dataset setting by introducing a novel Dataset-Aware Mixture-of-Experts, DAMEX layer that learns to make each expert an *'expert'* of its corresponding dataset while pooling shared information between the datasets in the rest of the network.

## 3 Preliminaries: Mixture-of-Experts (MoE)

An MoE layer consists of two parts: (1) a set of experts distributed across different GPUs, and (2) a gating function (router). The distribution across GPUs accelerate the inference speed of MoE while keeping the parameter count same. The gating function or router determines the destination GPU of each input token, then the tokens are *dispatched* to their target expert. These processed token representations are then *combined* such that the tokens are returned to their original location.

### 3.1 Routing of tokens

Let us denote the input tokens with $\mathbf{x} \in \mathcal{R}^D$, a set $E$ of experts by $\{e_i\}_{i=1}^{|E|}$ and router variable with $\mathbf{W_r} \in \mathcal{R}^{E \times D}$. The probability of each expert $p_i$ will then become:

$$g_x = \mathbf{W_r} \cdot \mathbf{x} \tag{1}$$

$$p_i(x) = \frac{\exp{(g_{x_i})}}{\sum_{j=1}^{|E|} \exp{(g_{x_j})}}. \tag{2}$$

After calculating the probability of each expert $p_i(x)$, [31] used *top-k* experts for routing the token. [31] calculates the output $y$ as a weighted combination of the processed tokens with their selection probability as:

$$y = \sum_{i \in \text{top-k}} p_i(x) e_i(x) \tag{3}$$

We use $k = 1$ as it results in a marginal degradation in performance but a huge gain in throughput [6] and [12]. The output $y$ is then passed forward to the remaining layers of the neural net.

### 3.2 Load balancing among the experts

Sparsity in the activation of experts is desirable for the MoE layer with respect to a given token. A simplistic approach would be to select the *top-k* experts based on the *softmax* probability distribution. In practice however, this approach causes majority of the tokens to be assigned to a small number of busy experts causing a load imbalance during training. This leads to a significant input buffer for a few busy experts, causing a delay in the training process, while other experts remain untrained [6]. Additionally, many experts may not receive adequate training. Therefore, a better gating function design is needed to distribute the processing burden equally across all experts.

To ensure better distribution of tokens, an auxiliary loss on experts is added along with the main task loss [18, 12]. The load balancing auxiliary loss $\mathcal{L}_{\text{load-balancing}}$ can be written as an average of $\mathcal{L}_{\text{importance}}$ and $\mathcal{L}_{\text{load}}$.

The importance loss $\mathcal{L}_{\text{importance}}$ for expert $e_i$ is calculated by

$$I_i = \sum_{x \in \text{Im}} p_i(x) \tag{4}$$

$$\mathcal{L}_{\text{importance}} = \frac{\text{Var}(I)}{\text{Mean}(I)^2} \tag{5}$$

Let us denote the normal distribution as $\mathcal{N}(\mathbf{0}, \sigma^2 \mathbf{I})$, where $\sigma = \frac{\text{gate noise}}{|E|}$. The CDF of this normal distribution is denoted as $\Phi(.)$. For load $L_i$ for expert $e_i$, we sample the CDF at probabilities $p_i(x)$

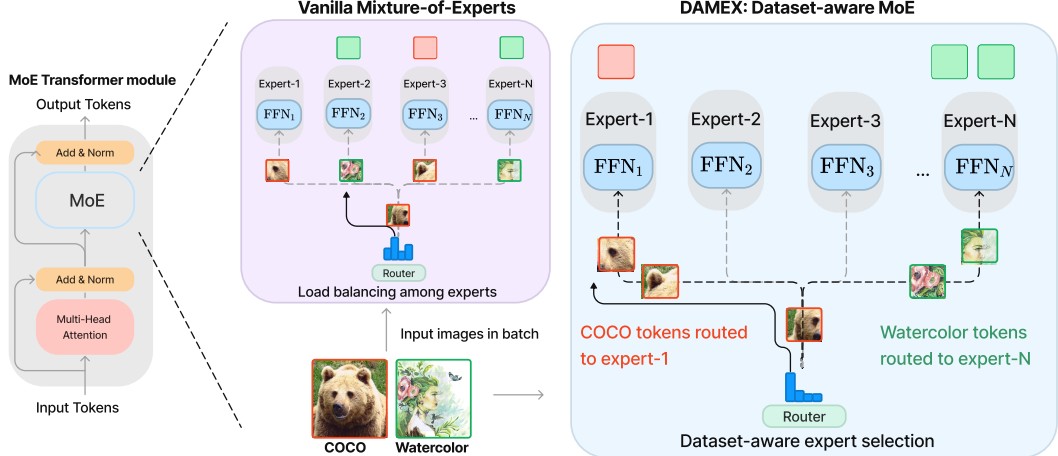

Figure 2: **Overview of DAMEX**: The dense FFN of a transformer block is replaced by an MoE layer. We show the difference between functioning of vanilla MoE layer vs DAMEX. Given two images from COCO and Watercolor dataset, the vanilla MoE router tends to balance the distribution of input tokens to each expert while DAMEX learns to route dataset tokens to their corresponding expert (Here expert-1 is assigned to COCO dataset and expert-N to Watercolor dataset). During inference, the trained router choose the appropriate expert depending on the input token without any information of dataset source.

as $\Phi(p_i(x))$ and sum it over all the tokens $x$ in the image Im. The load $L_i$ denotes the number of assignments among the experts. The loss $\mathcal{L}_{\text{load}}$ becomes

$$L_i = \sum_{x \in \text{Im}} \Phi(p_i(x)) \tag{6}$$

$$\mathcal{L}_{\text{load}} = \frac{\text{Var}(L)}{\text{Mean}(L)^2}. \tag{7}$$

Finally, the load balancing auxiliary loss $\mathcal{L}_{\text{load-balancing}}$ is calculated by

$$\mathcal{L}_{\text{load-balancing}} = \frac{L_{\text{importance}} + L_{\text{load}}}{2}. \tag{8}$$

## 4 Method

Our method Dataset-aware Mixture-of-Experts, DAMEX is based on Mixture of Experts (MoE) applied over transformer-based object detection pipeline DINO (DETR with Improved denoising anchor boxes) [42] such that each expert is trained to align to a corresponding dataset, subtly making it an actual 'expert' of that dataset-specific features. In this section, we will explain our method in detail.

### 4.1 Setup

We replace alternate non-MoE transformer modules from decoder of DINO architecture with MoE transformer modules. A common problem that occurs with mixing of datasets in object-detection is to have separate detection heads for each dataset which increases the number of parameters significantly in a two-stage detection pipeline [35, 45]. We leverage the transformer architecture of DINO in handling the number of classes by increasing the dimension of the last classification layer which results in marginal increase in number of parameters ($\approx 10,000$) compared to total model parameters, $46M$. In all experiments, we use one expert per GPU which keeps our parameter count same as that of non-MoE method (with an addition of a tiny router linear layer).

In vanilla MoE unlike image classification problem where every visual token is used for loss calculation [12, 2, 29], we apply the load-balancing loss Equation 8 only on the foreground tokens in the detection task. We find that the foreground-only loss balancing results in better gradient updates and mitigate expert representation collapse on detection task.

## 4.2   DAMEX: Dataset-aware Mixture-of-Experts

Vanilla MoE balances the experts usage across GPUs to utilize maximum benefit from ensembling features from input tokens. However, it can lead to inefficiency in knowledge sharing among experts due to distribution shift. For *e.g.* , the expert domain knowledge required for a natural images dataset like COCO is very different from a surveillance-style dataset like DOTA and we would want to learn their dataset-specific characteristics. Thus, to solve this inefficiency with using MoE on a mixture-of-datasets we introduce Dataset-aware Mixture-of-Experts (DAMEX) where we train the MoE router to route tokens depending on their dataset origin.

Given a set $D$ of datasets $\{d_m\}_{m=1}^{|D|}$, such that input token $\mathbf{x} \in d_m$. We define a mapping function $h : \{D\} \to \{E\}$ such that each dataset $d_m$ is assigned a specific expert $e_i$, then auxiliary loss $\mathcal{L}_{\text{DAMEX}}$ is calculated by a cross-entropy loss between *logits* $p_i$ (probability of selection of expert $e_i$) and labels $h(d_m)$ (target expert for token $x$ from dataset $d_m$).

$$\mathcal{L}_{\text{DAMEX}} = -\sum_{i=1}^{|E|} \mathbb{1}(h(d_m) = i) \log p_i(\mathbf{x}) \tag{9}$$

DAMEX trains the MoE router to dispatch all the visual tokens of the dataset to its corresponding expert as shown in Figure 2. Selecting specific expert ensure efficient usage of MoE and in-turn avoid their representation collapse.

## 5   Experiments

**Datasets.**   We evaluate the presented approach on Universal Object Detection Benchmark (UODB) [35]. UODB comprises of 11 datasets: Pascal VOC [5], WiderFace [40], KITTI [8], LISA [26], DOTA [36], COCO [22], Watercolor, Clipart, Comic [13], Kitchen [9] and DeepLesions [38], shown in Figure 1. Pascal VOC and COCO have natural everyday object images while Kitchen features indoor kitchen items. WiderFace is a huge human-face web images dataset covering a large variety of regions across the globe. Both KITTI and LISA datasets are based on traffic scenes captured with cameras mounted on moving vehicles. KITTI includes categories such as vehicles, pedestrians, and cyclists, whereas LISA is primarily composed of traffic signs. DOTA is a surveillance dataset with aerial images annotating planes, ships, harbors *etc.*. Watercolor, Clipart and Comic are cross-domain datasets in their separate styles of images. Comic have the same label set as Pascal VOC, while Watercolor and Clipart are a subset of that label set. DeepLesions is a medical CT images dataset highlighting lesions. Altogether, the UODB offers a wide array of domains, label sets, camera views and image styles, providing a comprehensive suite for evaluating multi-dataset object detection models.

**Implementation details.**   Following recent end-to-end object-detection pipelines [23, 19] we adapted MoE on current state-of-the-art, DINO [42]. We use pre-trained ImageNet ResNet-50 backbone with 4 scale features as input across all methods. For hyper-parameters, as in DINO, we use a 6-layer Transformer encoder and a 6-layer Transformer decoder and 256 as the hidden feature dimension. We altered DINO's decoder with alternate MoE layers following [12] using TUTEL library [12]. We use a capacity factor $f$ of 1.25 and an auxiliary expert-balancing loss weight of 0.1 with *top-1* selection of experts. We kept one expert per GPU and train on 8 RTX6000 GPUs with a batch-size of 2 per GPU, unless mentioned otherwise. Note that the number of parameters of model remains the same (except the addition of marginal router parameters) as each expert reside on a separate GPU replacing the existing feed-forward layer. For DAMEX, we keep the 1:1 dataset-mapping by assigning single expert to each dataset, except for full UODB run where we assign a single expert to COCO, VOC and Clipart dataset while KITTI and LISA shared a expert to fit 8 experts (on 8-GPU machine) on 11 datasets. We use a learning-rate of $1.4e\text{-}4$ and kept other DINO-specific hyperparameters same as [42].

Table 1: **Multi-dataset object detection results on UODB benchmark:** We report mean AP score on UODB set. Mixing refers to whether the model is trained with all the datasets mixed together or individually. The top two rows denote single domain models which are separately trained on each set. Observe that MoE hardly increases the number of parameters but show a jump in performance.

| Method | Mixing | Params | KITTI | VOC | WiderFace | LISA | Kitchen | COCO | DOTA | DeepLesion | Comic | Clipart | Watercolor | Mean |
|---|---|---|---|---|---|---|---|---|---|---|---|---|---|---|
| DINO [42] | - | 46.67M x 11 | **68.8** | 57.6 | 35.2 | 78.8 | 46.4 | 43.1 | 40.7 | 39.6 | 11.1 | 10.1 | 14.8 | 40.6 |
| Single dataset DINO-MoE | - | 46.68M x 11 | 68.2 | **58.0** | **35.3** | **79.2** | 47.0 | **43.4** | 41.0 | 40.6 | 8.4 | 10.0 | 14.8 | 40.5 |
| Wang *et al.* [35] | ✓ | 44.75M | 21.5 | 51.4 | 23.0 | 59.2 | 49.8 | 28.5 | 30.1 | 26.0 | **25.7** | 29.7 | **30.7** | 34.1 |
| Zhou *et al.*[45] | ✓ | 70.18M | 47.5 | 30.7 | 26.3 | 60.7 | 42.6 | 16.5 | 28.8 | 22.4 | 20.4 | 24.4 | 21.2 | 31.1 |
| DINO | ✓ | 46.73M | 58.4 | 53.9 | 34.7 | 73.0 | 48.4 | 39.9 | 44.7 | 42.9 | 21.2 | 27.8 | 20.1 | 42.3 |
| DINO-MoE | ✓ | 46.74M | 57.9 | 56.4 | 35.0 | 74.0 | **49.8** | 40.2 | 45.2 | 43.0 | 24.1 | 27.3 | 19.9 | 43.0 |
| DAMEX(Ours) | ✓ | 46.74M | 61.8 | 56.0 | 35.1 | 74.9 | 49.4 | 41.3 | **46.5** | **43.3** | 25.4 | **29.7** | 23.5 | **44.3** |

**Baselines and metrics.** We compare our approach against [35] that is the previous *state-of-the-art* on UODB benchmark. [35] gains a further advantage as during inference, the object detector knows from which dataset each test image comes from and thus makes predictions in the corresponding label space. On the other hand, our setup is more challenging as during inference the method is agnostic to the source dataset. We also compare our method against Partitioned detector of [45] as their work is complimentary to our approach and establish a universal detector with a common backbone. ResNet-50 is the common backbone across all the methods in this paper, including baselines. For fair comparison on the architectural gains, we prepare competitive baselines by running DINO on each dataset individually as well as mixing all the datasets together. All the reported numbers in this work are mean Average Precision (AP) scores evaluated on the available test or val set of corresponding dataset.

## 5.1   Multi-dataset Object Detection on UODB

In this section, we highlight our main results where we compare our DINO baseline against MoE DINO setup along with our proposed approach DAMEX on complete UODB datasets. Moreover, we re-evaluate the AP scores of [35] by running their best released model against COCO AP evaluation. Table 1 shows the AP scores on individual dataset and average performance on UODB. We observe that the proposed method improve accuracy by an average of +10.2 AP score with previous state-of-the-art and shows an average improvement of +2.0 AP on Mixed DINO baseline. Individually, we observe an increase in performance with DAMEX on out-of-domain datasets, like DOTA (surveillance aerial images) and DeepLesion (medical CT images) due to experts learning dataset specific features. While, KITTI suffer from a performance drop due to the reduced representation in the batch. Overall, dataset-aware MoE, DAMEX outperforms vanilla-MoE by +1.3 AP score highlighting the benefit of our proposed dataset-specific experts in multi-dataset training.

## 5.2   Analysis

We analyze the benefit of DAMEX in mixing datasets by quantitatively and qualitatively evaluating it for the desired properties associated with a universal object detector such as, limited data mixing, domain adaptation and same domain but different annotation set mixing. Further, we visualize the deeper routing decisions made by DAMEX and vanilla MoE across mixed dataset classes to highlight the efficiency and effectiveness of DAMEX.

**Limited data mixing.** In this setup, we demonstrate that the MoE based object detection model is better in handling imbalanced data. For this, we take KITTI and Kitchen dataset from the UODB set and create several few-shot scenarios by taking all the original images of Kitchen dataset and pair it against 50-shot, 100-shot, 1000-shot and complete KITTI dataset. The results have been presented in the Table 2. We demonstrate that the DAMEX is better in handling imbalanced even as we decrease the number of training images with a relative percentage gain of 15.7% in 50-shot and 24.3% in 100-shot setting over non-MoE baseline. We observe that the gap between DAMEX and vanilla MoE decreases with increase in data points. We believe that vanilla MoE is able to find a similar optimal distribution at the class-level with large amount of data.

**Domain adaptation.** Another common scenario of mixing dataset is when there are two datasets with the same annotation set but different domains. To replicate this setting, we conduct our analysis on a mixture of Watercolor and Comic dataset. Both Watercolor and Comic have the same label set

Table 2: **Limited Data Mixing:** We mix Kitchen dataset with n-shot examples of KITTI dataset. On right, we can observe the relative percentage gain of MoE and DAMEX over non-MoE method. In low-resource setting, DAMEX outperforms vanilla MoE due to efficient expert distribution whereas the gap diminishes as MoE found a similar distribution at the class-level with large amount of data.

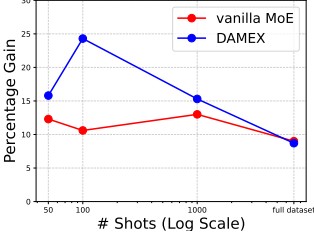

| # of examples | 50-shot | | 100-shot | | 1000-shot | | Full | |
|---|---|---|---|---|---|---|---|---|
| Method | KITTI | Kitchen | KITTI | Kitchen | KITTI | Kitchen | KITTI | Kitchen |
| Mixed DINO | 14.6 | 46.3 | 18.9 | 48.1 | 40.0 | 48.5 | 63.0 | 48.1 |
| Mixed DINO-MoE | 16.4 | 47.6 | 20.9 | 46.3 | 45.2 | 47.0 | **68.7** | 48.3 |
| DAMEX(Ours) | **16.9** | 47.7 | **23.5** | 46.0 | **46.1** | 47.5 | 68.5 | 48.3 |

but have very different styles of images. Watercolor has unique characteristics of watercolor art, such as transparent, fluid color washes, and visible brush strokes while Comic have bold, outlined objects and figures, with high-contrast color fills. The performance comparison of DAMEX, vanilla MoE, and non-MoE baseline is demonstrated in Table 3. We note a significant +2.9 AP increase on Watercolor and Comic datasets, underscoring that specialized experts can effectively learn dataset-specific traits, while shared network components can aggregate shared knowledge of the datasets.

Table 3: **Domain adaptation:** Watercolor and Comic both have same label sets but different style image domains. By learning dataset-specific features, DAMEX efficiently adapts to both dataset compared to vanilla MoE and non-MoE baseline.

| Method | Watercolor | Comic |
|---|---|---|
| Mixed DINO | 15.3 | 10.7 |
| Mixed DINO-MoE | 16.6 | 12.9 |
| DAMEX(Ours) | **18.2** | **13.6** |

Table 4: **Addressing Divergent Label Sets in Mixed Datasets:** KITTI and LISA, though both consist of traffic imagery, possess entirely disparate annotation sets, leading to potentially conflicting gradient updates. Nevertheless, DAMEX effectively circumvents this issue by learning dataset-specific features.

| Method | KITTI | LISA |
|---|---|---|
| Mixed DINO | 67.7 | 76.6 |
| Mixed DINO-MoE | 69.2 | 77.9 |
| DAMEX(Ours) | **69.4** | **78.5** |

Table 5: **Effect of dataset-expert mapping in DAMEX:** We compare the effect of dataset-expert mapping on UODB set. We observe that random assignment is still better than vanilla MoE which is trained on load-balancing loss while DAMEX experts learns dataset-specific features. Further, adding a human prior (or domain knowledge) in mapping can lead to further performance improvements in DAMEX.

| Method | KITTI | VOC | WiderFace | LISA | Kitchen | COCO | DOTA | DeepLesion | Comic | Clipart | Watercolor | Mean |
|---|---|---|---|---|---|---|---|---|---|---|---|---|
| Mixed DINO-MoE | 57.9 | 56.4 | 35.0 | 74.0 | 49.8 | 40.2 | 45.2 | 43.0 | 24.1 | 27.3 | 19.9 | 43.0 |
| DAMEX: Random mapping | 60.8 | 56.2 | 35.2 | 73.9 | 48.8 | 41.8 | 46.0 | 43.6 | 24.3 | 28.5 | 21.6 | 43.7 |
| DAMEX: Human-prior mapping (ours) | 61.8 | 56.0 | 35.1 | 74.9 | 49.4 | 41.3 | 46.5 | 43.3 | 25.4 | 29.7 | 23.5 | **44.3** |

**Mixing divergent label sets but within same domain.** Having two datasets within the same domain but having different annotation set is a common problem with mixing datasets. A universal detector should be robust against such missing labels scenario. We conduct our analysis on the mixture of KITTI and LISA datasets. Both are traffic images dataset, however while KITTI label set contains pedestrians, cars and cyclists, LISA label set contains only traffic signs such as speed limit, stop, no turn and warning sign. Note that these classes are present in both the datasets which make this a more challenging and realistic scenario for a universal detector. Table 4 depicts the performance of DAMEX against vanilla MoE and non-MoE baseline. We observe that dataset-aware MoE outperforms the baseline with +1.7 AP score on KITTI and +2.9 AP score on LISA dataset, highlighting that universal detector can benefit from dataset-specific experts.

**Deeper routing decisions.** We analyze the expert utilization of MoE layer in vanilla MoE against DAMEX. Figure 3 illustrates the distribution of expert selection weights over mixture of datasets. The plots were produced by running Mixed DINO-MoE and DAMEX from Table 1 on UODB set.

The datasets belonging to same domain are mapped to same expert. Figure 3 show that experts are equally distributed across datasets with shallow MoE layer suffering from poor expert utilization. Vanilla MoE fail to specialize in discriminating among datasets. DAMEX on the other hand efficiently use experts across datasets and conclusively learn to select appropriate dataset-specific expert during inference. The router selection improves in deeper MoE layers due to dataset-specific features learnt by earlier MoE layers. Moreover, DAMEX ensures fair expert utilization across all MoE layers.

**Effect of human-prior in dataset-expert mapping.**    A central part of DAMEX is the dataset-expert mapping $h : \{D\} \rightarrow \{E\}$ that the router learns during training. However, a question remains on how this mapping assignment affect the visual understanding of the model overall? We conduct an experiment where we randomly assign UODB datasets to experts and found a decrease in performance, as shown in Table 5. DAMEX allows the user to incorporate human-prior or domain knowledge in mapping datasets to experts. This in-turn helps in sharing similar features from larger datasets, *e.g.* COCO and VOC belong to the same domain, *natural images*. In our experience, assigning datasets with similar domains to same expert tend to help performance while keeping disparate domains to separate experts.

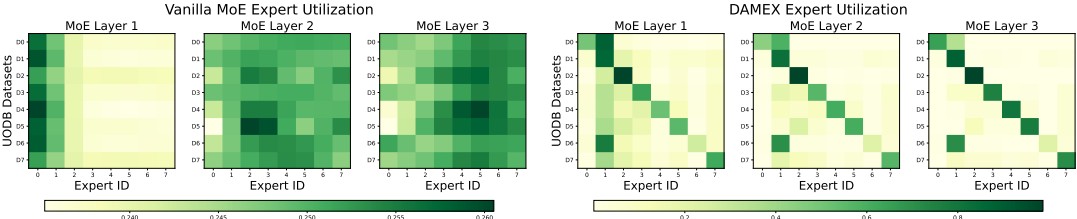

Figure 3: **Expert Utilization in DAMEX across datasets:** We show distribution of expert selection weights across dataset assignments. Here, D$n$ on $y$-axis refer to datasets mapped to expert ID $n$ while $x$-axis correspond to 8 experts. The (expert $e$, dataset $d$) pair denote the average routing weights of the tokens corresponding to bounding box of examples in dataset $d$. We can observe that DAMEX learns to route tokens to their mapped experts and is able to replicate it during inference without the knowledge of dataset-source. On the other hand, vanilla MoE struggles to efficiently utilize experts and suffer with mode collapse at Layer 1.

**Effect of Number of Experts on DAMEX.**    We study the effect of number of experts on DAMEX by experimenting on a mixture of four disparate domain datasets, namely KITTI, DOTA, VOC and Clipart to maximize the effect of dataset-aware experts. Table 6 shows the variation of mean AP score across the four datasets. We can observe that the best performance is obtained when number of experts are same as number of datasets. This observation is different

Table 6: **Effect of number of experts on DAMEX**

| # Experts | Clipart | DOTA | KITTI | VOC | Mean |
|---|---|---|---|---|---|
| 2 | 33.3 | 44.6 | 58.9 | 57.4 | 48.6 |
| 4 | **34.0** | **45.3** | **62.0** | **57.7** | **49.7** |
| 8 | 32.6 | 44.9 | 60.9 | 56.5 | 48.7 |

from vanilla MoEs where performance increases with increase in number of experts [12, 29]. It highlights that DAMEX is an efficient method for mixing datasets both in terms of compute and number of parameters as less number of experts require lesser compute. Note that for 8 experts, we are mapping two experts per dataset with equal probability during training while expert selection during inference is done by the trained router.

## 6    Conclusion

In this work, we have presented a novel Dataset-aware Mixture-of-Experts (DAMEX) layer for the object-detection task. We demonstrate the potential of MoE models for mixing datasets on object-detection task. The introduced DAMEX model leverages the MoE architecture to learn dataset-specific features, thereby enhancing expert utilization and significantly improving the performance on a universal object-detection benchmark with marginal increase in parameters. Our experimental results establish a new *state-of-the-art* performance on UODB by achieving an average increase of +10.2 AP score over the previous baseline and +2.0 AP score over non-MoE method.

# 7 Limitations & Social Impact

One limitation of our approach is that we are currently concatenating the classes of each dataset which results in a duplicate recognition of overlapping classes. A future work in unifying these labels over our setup can be an interesting avenue.

Going forward it is important that we ensure that the universal object detectors are unbiased and maintain fairness in their predictions. Here unbiased refers to classes with very few training examples or domain shift which can be called underrepresented classes. The detectors should be able to work on both classes with large, medium and few training samples in any environment. Similarly, for fairness, a universal object detector should be able to handle objects from different geographical regions, races, genders, etc. These are issues that we believe are ameliorated by methods focused on learning under imbalance or domain shift.

As a result, we feel DAMEX could be a method that takes a step towards this where each dataset, even with very few samples can be represented by different experts. We can potentially divide a large dataset into different sets which cater to at least one aspect of fairness and bias. However, we understand the need for further analysis and research before we have a truly unbiased and fair universal object detector and implore the community to do so too.

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
