# DAMEX: Dataset-aware Mixture-of-Experts for visual understanding of mixture-of-datasets

**Supplementary Material**

1 This document provides detailed derivation for the convergence behaviour of vanilla MoE on mixture-
2 of-datasets.

## 1 Mixing datasets in Vanilla MoE does not guarantee convergence with fixed number of experts

5 In the main paper, we empirically show the benefits of using our proposed DAMEX against vanilla
6 MoE when mixing multiple datasets. Here we provide theoretical evidence that vanilla MoE do not
7 guarantee convergence when mixing multiple datasets. Hence motivating the need for our method.

8 **Problem formulation.**  Consider a binary classification problem over $P$-patch inputs where each
9 patch has $d$ dimensions and label $y = \{\pm 1\}$. Thus, a labeled data point $(\mathbf{x}, y)$ has input $\mathbf{x} =$
10 $(\mathbf{x}^{(1)}, \mathbf{x}^{(2)}, \mathbf{x}^{(3)}, \ldots, \mathbf{x}^{(P)}) \in (\mathcal{R}^d)^P$ is a collection of P patch inputs with $y$ as the data label. The
11 data $\mathbf{x}$ is generated from $K$ clusters.

12 Chen et al. [2022] proves that in such a binary-classification problem, an MoE layer converges to an
13 $o(1)$ test loss and zero training loss. Since, such a classification problem has an intrinsic clustering
14 structure that may be utilized to achieve better performance. Examples can be divided into K clusters
15 $\bigcup_{k \in [K]} \Omega_k$ based on the distance from the cluster-center: an example $(\mathbf{x}, y) \in \Omega_k$ if and only if at
16 least one patch of $x$ aligns with cluster $\Omega_k$.

17 Specifically, Chen et al. [2022] Lemma 5.2 states that with data $\mathbf{x} \in (\mathcal{R}^d)^P$ the expert $m \in M_k$ can
18 achieve nearly zero test error on the cluster $\Omega_k$ but high test error on other clusters $\Omega_{k'}, k' \neq k$.

19 We will extend their formulation to multiple datasets $\mathbf{x}_1$ and $\mathbf{x}_2$ drawn from similar distribution.

20 **Lemma 1:**  A mixture of two datasets $\mathbf{x} = \mathbf{x}_1 \cup \mathbf{x}_2$, does not always results in the convergence of a
21 MoE layer, such that an expert $m \in [M]$ achieve nearly zero test error on cluster $\Omega_{k_1}$ and $\Omega_{k_2}$ but
22 high test error on all other clusters in both distributions $\Omega_{k'_1}, k'_1 \neq k_1$ and $\Omega_{k'_2}, k'_2 \neq k_2$.

23 **Proof:**  We will prove this by contradiction. Assume that we choose a $K$, such that VC dimension
24 of $[M]$ experts is equal to $K$. Since, $\mathbf{x}_2 \neq \mathbf{x}_1$, the VC dimension of a model required to learn both
25 $\mathbf{x}_1 \cup \mathbf{x}_2$ is at least $K + 1$. But, our $[M]$ experts cannot converge for more than $K$ VC dimension.
26 Contradiction.

27 Thus, we show that vanilla MoE does not guarantee convergence with mixture of datasets. However,
28 if we divide the dataset-expert pair using the proposed DAMEX approach then we can ensure that
29 each expert attends to a separate input data distribution $x$ leading to better convergence.

## References

[1] Zixiang Chen, Yihe Deng, Yue Wu, Quanquan Gu, and Yuanzhi Li. Towards understanding mixture of experts in deep learning. *arXiv preprint arXiv:2208.02813*, 2022.