# OpenReview forum: "DAMEX: Dataset-aware Mixture-of-Experts for visual understanding of mixture-of-datasets"
_NeurIPS.cc/2023/Conference — NeurIPS 2023 poster_

### Official Review · Reviewer_xzJh · 2023-06-20

**Soundness:** 2 fair
**Presentation:** 3 good
**Contribution:** 2 fair
**Rating:** 5
**Confidence:** 4

**Summary:**

This manuscript is motivated from training a universal object detector on the mixture of unlimited counts of datasets. Concretely, the previous works cannot scale the MoE due to budget, and the conventional MoE using balanced route suffers from knowledge sharing issues. To tackle the aforementioned issues, the authors devised a Dataset Aware Mixture of Experts (DAMEX), in which a novel MoE loss is proposed to learn the dataset index distribution. Then, extensive experiments are conducted on various settings to show the effectiveness.

**Strengths:**

1. The paper is well-organized and well-motivated. The authors provide detailed explaination to the conventional MoE and provides certain discussion on the motivation.
2. The idea of this manuscript is self-consistency, where the motivation is to tackle the knowledge distribution shift among datasets for existing MoE, and the $\mathcal{L}_{DAMEX}$ pushes router learning the dataset index distribution seems to be work in intuition.
3. The conducted experiments are wide and adequate to demonstrate the effectiveness of the proposed method.

**Weaknesses:**

1. The contribution of this manuscript is limited. All of the technical contributions are summarized as loss function of DAMEX, while it seems not settle the dataset distribution shift well. Concretely, as shown in Tab. 1, DAMEX fails to balance the results among datasets. As far as I am concerned, the dataset shift can be treated as hardness. To balance it, a larger improvement should be gained at harder datasets. Comparing with baseline DINO-MoE, the improvement of DAMEX concentrates on KITTI (3.9), COCO (1.1), DOTA (1.3), Clipart (2.4), Watercolor (3.6). There goes without strong correlation with dataset hardness. Welcome further discussion.
2. The manuscript has unfair expression. The authors claimed they outperform the state-of-the-art with almost 10% AP. I cannot agree with a paper published in 2019 could be called as sota, and the DINO looks to be sota.


**Questions:**

Why the authors repeatedly mention GPU? Is there any efficient design for DAMEX? How to implement MoE on multi-GPU has been mentioned in Sec. 3, is this necessary to keep mentioning GPU setting?

**Limitations:**

As discussed in Weakness, I don' t agree with the authors tackle the dataset shift. But this can be further discussed.

---

> ### Author Rebuttal · Authors · 2023-08-10
>
> We thank the reviewer for investing their time in reviewing our work and suggesting improvements.
>
>
> Q. “The contributions of …”
>
> We want to thank the reviewer for their feedback and wish to re-highlight our contributions.
>
> -	DAMEX does not require test-time dataset labels as DAMEX learns to route appropriate dataset-expert during training itself as mentioned in L226 under “Baselines and Metrics”. This is a more challenging setup compared to Wang et al. [1] or individual detectors which have test-time dataset source information.  Our observation is that by providing supervision over expert selection in DAMEX eliminates the need of dataset source during inference as each expert learns dataset-specific features.
>
> -	DAMEX avoids mode collapse of vanilla MoE by learning dataset-specific experts. Furthermore, DAMEX provides better representation learning by utilizing human-prior over dataset characteristics as a part of its dataset to expert mapping. This can be seen in Table A2 (rebuttal pdf) where DAMEX mapping is compared against a random assignment and vanilla MoE. We would like to extend our gratitude to R1 for suggesting this experiment and making our manuscript better.
>
> -	The number of parameters of DAMEX are smaller than baselines with individual detectors having 11x more parameters and previous works [2] having 1.5x more parameters. Yet, we observe consistent improvements from DAMEX on (1) UODB set, (2) Limited data availability, (3) Disparate domains, and (4) Divergent label sets.
>
>
> We note reviewer’s point of view regarding harder datasets and observe that DAMEX gains are distributed across multiple dataset domains, natural images, traffic images, aerial images and style images. Further, prior to this work MoE are considered to be a scalability tool only and DINO-MoE hasn’t been implemented before in this context. We believe that DINO-MoE is a strong baseline and is a part of our contribution in showcasing that MoE can handle mixture-of-datasets well. A fair comparison would be against DINO where we can observe large gains across the UODB set.
>
>
> Q. “The manuscript has …”
>
> Duly noted. We discuss this concern in the common answers and plan to update the paper accordingly.
>
> Q. “Why the authors …”
>
> Thanks for bringing this to our notice. As you already pointed out, the implementation is discussed in Sec.3. In the final draft, we will reduce the repeated mentioning of GPU.
>
> [1] Xudong Wang, Zhaowei Cai, Dashan Gao, and Nuno Vasconcelos. Towards universal object detection by domain attention. In Proceedings of the IEEE Conference on Computer Vision and Pattern Recognition, pages 7289–7298, 2019

---

### Official Review · Reviewer_962D · 2023-06-27

**Soundness:** 2 fair
**Presentation:** 3 good
**Contribution:** 3 good
**Rating:** 5
**Confidence:** 3

**Summary:**

The paper aims to develop a universal object detector that is applicable to a wide range of object detection datasets. For this aim, the authors proposes Dataset-aware Mixture of Experts (DAMEX). DAMEX is an extension of the vanilla Mixture of Experts layer to the multi-dataset scenario, in which each dataset is aimed to be routed to a specific expert. The authors also suggest replacing the fully connected layer in the transformer blocks by DAMEX layers. They incorporate such transformer blocks into DINO object detector. The experiments show that the approach improves upon previous SOTA as well as training a detector specifically for each dataset.

**Strengths:**

- The paper adapts MoEs to obtain universal object detection, which is intuitive.
- The approach does not increase the number of parameters significantly while obtaining a MoE for several datasets.
- The gains in terms of AP are significant over previous baseline on UODB benchmark. Also, the improvement over training a different DINO for each dataset is notable with ~4 AP. The method also outperforms using a Vanilla MoE by 1.3 AP.
- The analysis part is comprehensive and it shows that the method does work as expected. For example, the model learns to assign the experts to the datasets properly.
- Generally speaking, the paper is clearly written.

**Weaknesses:**

1. I think this task is closely related to open vocabulary object detection, on which there are already several different baseline methods. Such models are also evaluated on a set of multiple object detection datasets, e.g. using Object Detection in the Wild benchmark [A]. As an example method, GLIP [A] can easily be prompted given the union of the label spaces of each dataset in UODB benchmark. By considering recent advances, I'd expect at least an example of these models to be considered as a baseline.

2. The authors use UODB benchmark (compared to Wang et. al, 2019) to evaluate their method. This dataset normally uses Pascal VOC style AP. Here, the authors use COCO-style AP. That's why, the AP values are significantly low in this paper (compared to Wang et. al, 2019) owing to the choice of this evaluation measure. It took me a while to dig into this and see the difference in evaluation. I'd expect this to be very clear in the metrics section.

3. Please note that k index to define summation is not used in p_i or e_i. I'd recommend checking that equation.

[A] Grounded Language-Image Pre-training, CVPR 2022

**Questions:**

1. Do you think open vocabulary object detection methods are also valid baselines for this task? Specifically, why don't you use a model such as GLIP to compare your method against?

2. You mention that each GPU stores the weights of one expert. Following from this, do you pay specific attention to sampling the batch? What happens if all images (16 in total across all GPUs) come from the same dataset, in which case they should be sent to the same GPU, potentially causing memory issues?

3. The learning rate of the proposed method is tuned to be 1.414e-4. Did you tune the learning rate for each dataset while obtaining the single domain models (the first two lines in Table 1)? If so, how?

4. Why did you prefer using COCO-style AP instead of the performance measure of the benchmark that is Pascal VOC style AP?

**Limitations:**

Yes

---

> ### Author Rebuttal · Authors · 2023-08-10
>
> We thank the reviewer for their time in reviewing our work.
>
> Q. “I think this …”
>
> We note the review feedback and their suggestion for GLIP. We answer the question under common answers and report the results in Table A1 (rebuttal pdf).
>
> Q. “The authors use UODB…”
>
> We understand reviewer's concerns regarding the metrics used in our work and Wang et al.[2]. We consistently report COCO AP scores as a metric throughout the paper (L233 and L238) as it is an average over multiple IoU thresholds between 0.5 (coarse localization) and 0.95 (near-perfect localization). Pascal Style AP score is calculated over a fixed IoU threshold of 0.5.  Nevertheless, to remove any doubts we have provided Pascal Style AP score of DAMEX against all the baselines in Table A3 in the rebuttal pdf.
>
> Q. “Please note…”
>
> We thank the reviewer for catching this typo, we will fix it in the camera-ready version.
>
> Q. “Do you think …”
>
> We covered this question previously.
>
> Q. “You mention that …”
>
> We are grateful for the reviewer for bringing this insightful point. Since the datasets are extremely imbalanced with Watercolor having only 500 train images vs 23k images of DeepLesion, we found in our initial experiments that uniform sampling across training images worked best for overall performance. For e.g., sampling based on inverse dataset size hurts overall performance. However, we agree that there can be a better sampling strategy for UODB which can improve DAMEX performance further and is agnostic to our contributions.
>
> In the scenario mentioned, the method does not suffer from memory issues but a decrease in speed as all tokens are processed through a single expert. This bottleneck exists in vanilla MoE also and is an inherent feature of Mixture-of-Experts.
>
> Q. “The learning rate …”
>
> Duly noted. We change the learning to sqrt(2)*1e-4 following Krizhevsky’s [1] recommendation of learning rate dependence on batch size as we used a batch size of 2 during training. We followed a similar strategy for all baselines for fair comparison.
>
> Q. “Why did you prefer …”
>
> Answered previously.
>
>  We hope it will resolve all the concerns!
>
>
> [1] Krizhevsky, A., 2014. One weird trick for parallelizing convolutional neural networks. arXiv preprint arXiv:1404.5997.
> [2] Xudong Wang, Zhaowei Cai, Dashan Gao, and Nuno Vasconcelos. Towards universal object detection by domain attention. In Proceedings of the IEEE Conference on Computer Vision and Pattern Recognition, pages 7289–7298, 2019

---

> > ### Comment · Reviewer_962D · 2023-08-20
> > **Rebuttal acknowledged**
> >
> > Thanks for your time and effort to address my concerns. I am happy to keep my tendency for acceptance.

---

### Official Review · Reviewer_1BaQ · 2023-07-06

**Soundness:** 3 good
**Presentation:** 2 fair
**Contribution:** 2 fair
**Rating:** 6
**Confidence:** 4

**Summary:**

This work, motivated by the goal of developing a “universal detector,” seeks to understand how best to train a model across a large set of existing curated, labeled datasets that might differ in collection strategy, labeling standards, categories of interest, etc. They posit that the best strategy is to train dataset-specific features within a model that can be used to jointly predict on new images via mixture-of-experts, where dataset tokens are used to route images through the network based on the “expertise” of the learned features effectively and with many fewer parameters than prior multi-headed approaches. They show nice improvement using their method on the Universal Object-Detection Benchmark, and demonstrate that their method reduces representation collapse, a common issue with MoE training.

After reading the authors rebuttal and discussing with them, I will maintain my score of 6.

**Strengths:**

This work, which extends MoE-based dataset mixing to object detection, is differentiated from prior work using MoE for dataset mixing in image classification via the inclusion of specific per-dataset-expertise-routing layers within a larger shared-across-datasets architecture. They  distribute expertise via a load balancing loss (similar to prior work), which they adapt to the detection setting by applying it only to foreground tokens (reducing the influence of background on expertise routing).  They use explicit dataset tokens to route information to specific experts during training and inference, which requires train- and test-time dataset labels. DAMEX particularly seems to do well over vanilla MoE in a few-shot setting, as seen in the figure in Table 2.

**Weaknesses:**

The claims of performance improvements of 10% on average are slightly misleading, as there are probably lots of additional contributing factors that are not attributable to their method that led to prior reported numbers being lower. They see only a 2.3% improvement on average over vanilla DINO, and their contribution of a dataset-specific routing layer only improves performance over DINO-MoE by 1.3%. On many test sets their method underperforms by a significant margin. I would appreciate a more nuanced and representative claim, and some additional investigation and analysis as to why their dataset performs better on, e.g., DOTA, and much worse on Watercolor or LISA.

Since the dataset tokens are explicit, I believe the model would need to be completely retrained to add a single additional source of data, which is (perhaps prohibitively) inefficient. Maybe an extension of this work would be to enable add-on routing/expertise generation for new mixing datasets without end-to-end retraining across all datasets.

Nit:

I’m not sure the citation style matches the NeurIPS standard?

Grammar is inconsistent and frequently in error throughout, with too many small mistakes to reasonably capture in this review. I would recommend significant text editing to increase the readability of a camera-ready manuscript for this work.

**Questions:**

How would you add an additional dataset into the mix with this method? Would it require training from scratch?

It seems in figure 3 that despite the load balancing, expertise is heavily relying on “expert 1” for many of the classes. How is this related to the class distribution across datasets?

**Limitations:**

More nuanced evaluation and discussion of failure modes of their method on some datasets but high performance on others would help the reader gain intuition as to when this method would be worth implementing and applying. What about the individual dataset structure contributes to when this dataset-specific routing helps vs hurts?

I do appreciate the discussion of label unification as a current limitation.

The comment “Going forward it is important that we ensure that the universal object detector are unbiased and maintain fairness in their predictions” feels like a throwaway. How would this work possibly contribute to that or how could it be adapted to contribute to that goal? How is the current paradigm possibly entrenching bias?

---

> ### Author Rebuttal · Authors · 2023-08-10
>
> We thank the reviewer for their time in understanding our work and providing feedback to improve the manuscript.
>
> Q. “The claims of performance…”
> Duly noted. As mentioned in common concerns, we will change our writing to better convey the performance gains to the readers.
>
> We understand reviewer's concern regarding why some baselines are better on few datasets and we believe it is because:
>
> -  Individual detectors tend to overfit on datasets and have lot more parameters.
>
> - We choose DINO as our detection pipeline as it is SOTA in unimodal object-detection method. With DINO as base architecture, we observe that DAMEX improves further on the base numbers, and we see consistent gains across datasets with the best overall performance. Note that, DAMEX is an architecture agnostic idea of using MoE in a dataset-aware setting.
>
> - Individual detectors and Wang et al. have knowledge of dataset-source during test time which is absent for DAMEX (L105-107).
>
> We thank the reviewer for this insightful question and will update the writing to share these nuanced reasons.
>
> Q. “Since the dataset tokens…”
>
> Yes, this is a very good point by the reviewer. In our current setup the addition of a new dataset will require re-training from scratch. We agree that an extension of this work would be addition of a new dataset in a continual learning environment by introducing a new expert with some frozen layers in the shared network. We avoided delving into this scenario as it would have diluted our core idea of DAMEX. We motivate the community to pursue this as future work.
>
> Q. Question regarding Fig 3.
>
> We understand the issues with fig 3 and have cleared them up under common answers by providing a new figure (Figure A1, rebuttal pdf).
>
> Q. “What about the individual …”
>
> We agree with the reviewer's thinking. Individual dataset structure is important in assessing dataset-expert mapping. In our experience, assigning datasets with similar domains to same expert tend to help performance while keeping disparate domains to separate experts. We will add this in the camera-ready manuscript.
>
> Q. Nits
>
> We accept the change in citation style and will improve the grammar in the camera-ready manuscript. We will also make changes in Limitations sections as suggested by the reviewer.

---

> > ### Comment · Reviewer_1BaQ · 2023-08-19
> > **What about the potential implications of the work on bias and fairness, and mentioned in the text by the authors?**
> >
> > As I mentioned in my review, "The comment “Going forward it is important that we ensure that the universal object detector are unbiased and maintain fairness in their predictions” feels like a throwaway. How would this work possibly contribute to that or how could it be adapted to contribute to that goal? How is the current paradigm possibly entrenching bias?"
> >
> > I would appreciate if the authors respond to this comment, as I believe engaging more deeply with broader questions about the implications of our work is an important and necessary opportunity for growth in our field.
> >
> > I thank the authors for their thoughtful responses to my other questions and concerns.

---

> > > ### Author Response · Authors · 2023-08-20
> > >
> > > We thank the reviewer for their engagement and discussion.
> > >
> > > Universal object detectors should be unbiased and fair. Here unbiased refers to classes with very few training examples or domain shift which can be called underrepresented classes. The detectors should be able to work on both classes with large, medium and few training samples in any environment. Similarly, for fairness, a universal object detector should be able to handle objects from different geographical regions, races, genders, etc. These are issues that we believe are ameliorated by methods focused on learning under imbalance or domain shift.
> > >
> > > As a result, we feel DAMEX could be a method that takes a step towards this where each dataset, even with very few samples can be represented by different experts. We can potentially divide a large dataset into different sets which cater to at least one aspect of fairness and bias. However, we understand the need for further analysis and research before we have a truly unbiased and fair universal object detector and implore the community to do so too.

---

> > > > ### Comment · Reviewer_1BaQ · 2023-08-20
> > > >
> > > > Thank you for expanding on your thoughts on this important matter and clarifying how you feel DAMEX might contribute. I agree that additional analysis is needed to determine if and to what extent this is actually seen in practice. I know that space is limited, but I think trying to clarify the intent of the sentence currently included in the manuscript along these lines would be useful, as currently it is unclear.
> > > >
> > > > I very much appreciate the engagement and discussion, and all the additional insight provided by the authors. I will maintain my positive score for this work.

---

### Official Review · Reviewer_2x26 · 2023-07-06

**Soundness:** 2 fair
**Presentation:** 3 good
**Contribution:** 2 fair
**Rating:** 5
**Confidence:** 4

**Summary:**

This paper introduces a DAMEX layer based on the idea of assigning samples conforming to the charactersitiscs of a dataset to the corresponding expert. The underlying thought is to build dataset-relevant modules and ensemble them all together. Previous approaches leverage Mixture of Experts to scale their model while maintaining approximately the same inference speed. This work deals with the scenario of training multiple datasets together for object detection.

**Strengths:**

The writing is clear. Results seem to prove the effectiveness of the proposed approach. The added number of parameters is small.

**Weaknesses:**

1. There is no introduced novelty in the proposed approach. It is good to know dataset specific experts are good for improving performances on object detection but the proposed DAMEX is in principle a linear routing layer that maps the input to different experts. The training procedure is also very straightforward: training different datasets on different GPUs. Essentially, this is equivalent to model ensemble a per-block router. Most of credits should go to GShard and Tutel.
2. More clarification is needed for the improvement of performances. Throughout the results part, clarifications of why the proposed method is better than the others are missing. For example, I miss why DAMEX could deal with imbalanced datasets? Why it has better domain adaptation than DINO?
3. Scalability is limited. From the experimental setting and the results, the maximum number of experts is 8, which is pretty small. It seems that it is highly correlated with the number of GPUs as well. From Tab. 5, we can see that when multiple experts are distributed to a single dataset, the performance drop is significant, which demonstrates that the approach is not scalable.
4. Important experimental details are missing. For example, in Tab. 5, which two datasets are distributed together when # Experts = 2? The number of iterations needed for training is not given.
5. From the visualization in Fig. 3, DAMEX has more condensed cluster such as Expert 1. This may indicate that other experts are not exploited well and it might work to delete some experts with no obvious decrease on performance. This potentially would have robustness issues. Again, it is more like doing majority voting from the ensemble perspective.

**Questions:**

See weaknesses please.

**Limitations:**

No. The authors mentioned the negative societal impact but does not have a solution yet.

---

> ### Author Rebuttal · Authors · 2023-08-10
>
> We thank the reviewers for their valuable feedback.
>
> Q. “There is no introduced novelty…”
>
> We believe that performance gain and novelty are inter-related.
>
> Sparsely activated networks are composed of components (experts), each of which learn to handle a subset of the complete set of training cases.
> The key here is the division of training set into different subsets, or in other words, routing to different experts.
>
> GShard [1], Tutel [2] and Switch [3] learn the expert division at train time.
> We take a different approach. We utilize the extra information that is available in the form of dataset/domain to guide the division. The training involves letting one expert handle samples from one data set.
>
> -	As noted above, the idea is different from what is being done commonly in the form of MoEs. And thus, it is novel in the context of this problem.
>
> -	While it seems simple from the outlook, it is quite perplexing to see why it works so well, as noted by the reviewer.
>
> -	The fascinating part is that no such dataset label is provided at test time. But our training enables the network to recognize the dataset/domain. Thus, our novel loss function and model design allows the network to aggregate the knowledge across datasets in the shared parts and learn dataset-specific features in the dataset experts.
>
> - The dataset-expert mapping adds human-prior in the expert training which further helps performance. This can be seen in Table A2 (rebuttal pdf)  where DAMEX mapping is compared against a random assignment and vanilla MoE. We would like to extend our gratitude to R1 for suggesting this experiment and making our manuscript better.
>
> -	To summarize, this seemingly simple form of work division allows us to learn a good multi-dataset object detector. This finding is very useful for the community. This could lead the community to explore other kinds of expert division rather than just learning them at train time.
>
> Q. “More clarification is needed…”
>
> Thank you for your feedback. DAMEX is better than baselines in domain-adaptation and imbalanced datasets because dataset-aware experts are able to learn dataset-specific features while the shared part of the network aggregate common information from the datasets. This again connects to our contributions: (a) human-prior expert-dataset mapping, (b) dataset-specific experts training, (c) correct expert selection during inference. We will further improve our writing in the corresponding sections of the manuscript to emphasize these points.
>
> Q. “Scalability is limited…”
>
> We thank the reviewer for their question. We believe scalability can be achieved from two perspectives (a) Compute and (b) Datasets. We focus on the latter as the focus of our work. In DAMEX we know the number of datasets at training, hence, we can scale the number of optimal experts by matching it with the datasets. Instead of having 1 expert/gpu we can have 2 experts/gpu to accommodate higher expert load on the same compute. However, if we have less training datasets (say 4) then as shown in Tab 5, the optimal setup is using same number of experts as datasets. Having more experts (say 8, Tab 5) does not help in performance.
>
> Q. “Important experimental …”
>
> Duly noted. In Tab 5, VOC and KITTI share the same expert following same strategy as Tab 1. Other hyperparameters including number of epochs were set to be same as DINO (36 epochs) as mentioned at L223. We accept reviewer’s suggestions in writing and will clarify it in the camera-ready version.
>
> Q. “From the visualization …”
>
> We understand reviewer's concern for Fig 3 and have clarified it in common answers. We provide an updated Fig 3 as Fig A1 in rebuttal pdf.
>
> [1] Dmitry Lepikhin, HyoukJoong Lee, Yuanzhong Xu, Dehao Chen, Orhan Firat, Yanping Huang, Maxim Krikun, Noam Shazeer, and Zhifeng Chen. Gshard: Scaling giant models with condi tional computation and automatic sharding. arXiv preprint arXiv:2006.16668, 2020.
>
> [2] Changho Hwang, Wei Cui, Yifan Xiong, Ziyue Yang, Ze Liu, Han Hu, Zilong Wang, Rafael Salas, Jithin Jose, Prabhat Ram, Joe Chau, Peng Cheng, Fan Yang, Mao Yang, and Yongqiang Xiong. Tutel: Adaptive mixture-of-experts at scale. CoRR, abs/2206.03382, June 2022.
>
> [3] William Fedus, Barret Zoph, and Noam Shazeer. Switch transformers: Scaling to trillion parameter models with simple and efficient sparsity. The Journal of Machine Learning Research, 23(1):5232–5270, 2022.

---

> > ### Comment · Reviewer_2x26 · 2023-08-20
> > **Post rebuttal**
> >
> > I've read the authors' response. The author response addresses most of my concerns. I'm happy to increase my score.

---

### Official Review · Reviewer_UzKA · 2023-07-07

**Soundness:** 3 good
**Presentation:** 3 good
**Contribution:** 3 good
**Rating:** 6
**Confidence:** 3

**Summary:**

This paper tackles the problem of mixture-of-datasets training for object detection. The authors propose a mixture-of-expert-based (MoE) model that utilizes dataset-specific features to tackle mixing of heterogeneous datasets. The backbone is based on DINO transformer and one expert is assigned to one dataset. Results on the UODB dataset shows the efficacy of the proposed method.

**Strengths:**

1. The idea is straightforward and the intuition of using one expert per dataset makes sense. The method seems easy to implement and the authors promise to release code so the reproducibility is good.
2. The proposed method is better than the previous SOTA by a large margin.

**Weaknesses:**

1. The design of assigning one expert for each dataset seems limited. The MoE model learns to route the tokens to a specific dataset expert, but in the real world there are many more than 11 datasets/styles. For images that are watercolor-comic etc., would the model get confused? Also, please discuss the need for a universal object detector trained on limited supervised datasets when large vision-language foundation models (VLMs) like [4*] can already get 60+ mAP (vs. 41 in this paper) on COCO. It would help to see the performance of the VLMs on these lesser-tested datasets like DOTA and Watercolor.
2. The performance on common object detection benchmarks (MSCOCO, VOC, etc.) is lower than baselines. This may suggest that some features are difficult to learn under the MoE setting.

**Questions:**

1. The line of work on mixture-of-datasets for video understanding should also be discussed [1*, 2*, 3*] in related work. Also, open-vocabulary object detection based on large vision-language models (VLMs) [4*] should also be considered.
2. A inference speed comparison is needed. What is the FPS, for example, of the proposed method vs. others?
3. Other minor comments:
The citation format makes some sentences strange: “Firstly, these datasets have been collected over time Everingham et al. [2015] and …” (Line 29)
The text in Figure 2 is too small to read.

[1*] Akbari, H., Yuan, L., Qian, R., Chuang, W. H., Chang, S. F., Cui, Y., & Gong, B. (2021). Vatt: Transformers for multimodal self-supervised learning from raw video, audio and text. Advances in Neural Information Processing Systems, 34, 24206-24221.
[2*] Duan, H., Zhao, Y., Xiong, Y., Liu, W., & Lin, D. (2020, August). Omni-sourced webly-supervised learning for video recognition. In European Conference on Computer Vision (pp. 670-688). Cham: Springer International Publishing.
[3*] Liang, J., Zhang, E., Zhang, J., & Shen, C. (2022). Multi-dataset Training of Transformers for Robust Action Recognition. Advances in Neural Information Processing Systems, 35, 14475-14488.
[4*] Yuan, L., Chen, D., Chen, Y. L., Codella, N., Dai, X., Gao, J., ... & Zhang, P. (2021). Florence: A new foundation model for computer vision. arXiv preprint arXiv:2111.11432.
--------------------------Post rebuttal

I have read the author rebuttal and other reviewers. The authors have addressed my concerns and questions. I'm increasing my score.

**Limitations:**

The authors have discussed the limitation and potential issues of the research.

---

> ### Author Rebuttal · Authors · 2023-08-10
>
> We thank the reviewer for their feedback and investing their time in understanding our work.
>
> Q. “The design of assigning…”.
>
> We would like to thank the reviewer for this question that prompted us to run a new experiment and improve our manuscript further. DAMEX allows the user to incorporate human prior in mapping datasets to experts which in-turn helps in leveraging similar features from larger datasets. Table A2 (rebuttal pdf) shows that DAMEX w/ random expert-dataset mapping performs worse than DAMEX w/ human-prior mapping but is still better than learned experts through load-balancing loss as in vanilla MoE. This shows that mapping plays an important part in the performance and having a human-prior in mapping can boost model performance. Further, DAMEX isn’t limited to one dataset/expert as we show in Table 1, where Expert 1 shares COCO, VOC and Clipart dataset. We will add a section on mapping in the camera-ready manuscript based on these results.
>
> Q. “For images that are watercolor-comic etc.,...?”
>
> For the images that are watercolor-comic, we believe it will be routed to any of the watercolor or comic expert and should get equally good representations from both experts.
>
> Q. Need for VLFM.
>
> Thank you for the question. We have answered this question under common answers.
>
> Q. “A inference speed …”
>
> Thank you for the question. We find that the inference speed of DAMEX (2.99 FPS) is very similar to MoE (3.46) and believe that it can further be improved through code optimizations as the goal of our work is not higher inference speed but better performing mixture-of-datasets object detector.  Also, our implementation library (Tutel) is catered towards vanilla MoE not DAMEX.
>
>
> Q. “The performance on common…”
>
> Yes, we agree with the reviewer in pointing this out. We observe that single dataset detectors of COCO, VOC outperform DAMEX. This is a trend that we notice in all baseline dataset-mixing papers too [1,2] which have non-MoE architectures as well. Our understanding is that natural image dataset distribution is significantly impacted with other dataset distribution types due to lighting, camera views and scale.  Further, most of the parameters are shared between the datasets and there is only a fraction of dataset-specific parameters,
>
> Q. Writing suggestions.
>
> Thanks for the suggestions. As suggested, we will add a section on mixture-of-datasets for video understanding in related works. We acknowledge the issue with the citation style and fig 2. font size. We will fix the manuscript and incorporate the changes.
>
>
> [1] Xudong Wang, Zhaowei Cai, Dashan Gao, and Nuno Vasconcelos. Towards universal object detection by domain attention. In Proceedings of the IEEE Conference on Computer Vision and Pattern Recognition, pages 7289–7298, 2019
>
> [2] Xingyi Zhou, Vladlen Koltun, and Philipp Krähenbühl. Simple multi-dataset detection. In Proceedings of the IEEE/CVF Conference on Computer Vision and Pattern Recognition, pages 7571–7580, 2022.

---

### Author Rebuttal · Authors · 2023-08-09

We thank the reviewers for their feedback on DAMEX. Through their comments, we have gained a valuable insight of the paper from a reader’s perspective, and we are thankful for their suggestions on improving our manuscript.

1. **Comparison against Vision-Language Foundation models (R1, R4)**

Our method and baselines are based on vision-modality and that’s why we didn’t  compare it against VLFMs as has been done by recently published prior works [1,2]. However, following request of R1,R4, we ran zero-shot GLIP [3] on UODB benchmark and report the performance in Table A1 (rebuttal pdf). GLIP [3] has been pre-trained on large datasets and show very good performance on natural image datasets like VOC and COCO but fail to perform on other domains, notably DeepLesion (medical dataset). Finally, we want to highlight that DAMEX is an architectural solution for mixing datasets and can be also applied on VLFMs like GLIP that can be explored as a future work.

2. **Explanation of Fig 3 (R2, R3)**

Fig 3 indicates the expert selection with respect to all the classes in the mixture of 11 datasets in UODB. As mentioned in the caption of Fig 3, the majority of classes are from VOC and COCO dataset which are mapped to Expert 1 and hence, there is a higher expert 1 utilization as both of these datasets are majority in the set. Contrary to this in vanilla MoE, we can observe that MoE Layer 1 collapses for Expert 0 and 1 for all the datasets. However, we thank the reviewers for their suggestions, and we have realized that dividing expert utilization wrt classes has resulted in a confusing figure. Please refer to Figure A1 (rebuttal pdf), where we have updated the figure 3.  We compare datasets against expert IDs and observe high expert utilization for each dataset against its assigned expert.

3. **Reporting of improvement over prev. SOTA (R3, R5)**

We try to be transparent in our performance gains by introducing DINO based baselines for comparison of DAMEX. However, we understand reviewer’s concerns and will rephrase writing to help the readers understand our performance gains accurately.

[1] Hao Zhang, Feng Li, Shilong Liu, Lei Zhang, Hang Su, Jun Zhu, Lionel M Ni, and Heung-Yeung Shum. Dino: Detr with improved denoising anchor boxes for end-to-end object detection. arXiv preprint arXiv:2203.03605, 2022.

[2] Xingyi Zhou, Vladlen Koltun, and Philipp Krähenbühl. Simple multi-dataset detection. In Proceedings of the IEEE/CVF Conference on Computer Vision and Pattern Recognition, pages 7571–7580, 2022.

[3] Liunian Harold Li*, Pengchuan Zhang*, Haotian Zhang*, Jianwei Yang, Chunyuan Li, Yiwu Zhong, Lijuan Wang, Lu Yuan, Lei Zhang, Jenq-Neng Hwang, Kai-Wei Chang, and Jianfeng Gao. Grounded language-image pre-training. In CVPR, 2022

---

### Decision · Program_Chairs · 2023-09-21

**Decision:**

Accept (poster)

**Comment:**

Summary
This paper proposes a way to train object detection systems with a mixture of datasets. The authors use a mixture of experts that learns dataset specific features to tackle different datasets in detection. The method is evaluated on the Universal Object Detection benchmark.

Strengths
- The authors propose to use MoE for object detection.
- The idea of using a simple load balancing loss to route foreground tokens is elegant. This provides major improvement over a vanilla MoE.
- Proposed method is parameter efficient

Reviews & Justification
The paper received positive reviews overall. The proposed idea is simple and shows good empirical gains on UODB. The authors are encouraged to include the comments from the reviewers about positioning this work wrt open vocabulary detection, and explaining the gain in performance clearly.